# Novel Genetic Variants and Clinical Profiles in Peters Anomaly Spectrum Disorders

**DOI:** 10.3390/ijms26136454

**Published:** 2025-07-04

**Authors:** Flora Delas, Samuel Koller, Jordi Maggi, Alessandro Maspoli, Lisa Kurmann, Elena Lang, Wolfgang Berger, Christina Gerth-Kahlert

**Affiliations:** 1Institute of Medical Molecular Genetics, University of Zurich, 8952 Schlieren, Switzerland; 2Department of Ophthalmology, University Hospital of Zurich, 8091 Zurich, Switzerland; 3Neuroscience Center Zürich (ZNZ), University and ETH Zurich, 8006 Zurich, Switzerland; 4Zurich Center for Integrative Human Physiology (ZIHP), University of Zurich, 8006 Zurich, Switzerland

**Keywords:** Peters anomaly, Peters plus syndrome, Peters plus-like syndrome, corneal opacity, anterior segment dysgenesis, *FOXC1*, *PEX2*, *ZFHX4*, congenital glaucoma

## Abstract

Peters anomaly (PA) is a rare congenital disorder within the anterior segment dysgenesis (ASD) spectrum, characterized by corneal opacity, iridocorneal adhesions, and potential systemic involvement. The genetic basis of PA and related syndromes are complex and incompletely understood. This study investigates novel genetic variants and their clinical impact in two unrelated individuals diagnosed with PA spectrum disorder. Whole-exome sequencing (WES), long-range PCR, and breakpoint analysis were applied to identify pathogenic variants. In the first patient, a heterozygous ~1.6 Mb deletion was detected, spanning the genes *PEX2* and *ZFHX4* (GRCh37 chr8:g.76760782_78342600del). The second patient carried a heterozygous *FOXC1* variant (NM_001453.3:c.310A>G), classified as likely pathogenic. Both variants were confirmed by Sanger sequencing and considered de novo, as they were not present in the biological parents. Clinical evaluations revealed phenotypic variability, with the first patient displaying both ocular and systemic anomalies as in a Peters plus-like syndrome phenotype, while the second patient had isolated ocular manifestations as in a PA type 1 phenotype. These findings expand the genetic landscape of PA, underscoring the importance of comprehensive genomic analysis in subclassifying ASD disorders. Further studies are needed to elucidate the functional consequences of these variants and improve diagnostic and therapeutic strategies.

## 1. Introduction

Anterior segment dysgenesis (ASD) disorders encompass a broad spectrum of developmental abnormalities impacting the cornea, iris, and lens of the human eye [1]. Among these, Peters anomaly (PA) is a distinct subtype, characterized by variable corneal opacities and defects in the posterior layers of the cornea, with or without adhesions (synechiae) between the lens and/or iris and the cornea [1,2,3]. The prevalence of PA is estimated to be between 2.2 and 3.1 cases per 100,000 births [4]. First described by Alfred Peters in 1906, PA presented as a complex condition involving a shallow anterior chamber (AC), iridocorneal adhesions, central corneal leukoma, and Descemet’s membrane defects [3]. Four overlapping PA spectrum disease subtypes have been described in the literature (Table 1): type 1 (PA1), type 2 (PA2), Peter plus syndrome (PPS), and Peter plus-like syndrome (PPLS).

PA1 and PA2 were the initial sole classifications in PA spectrum disorder. PA1 arises from an incomplete disjunction of the cornea and iris, resulting in a mild-to-moderate degree of corneal opacity [2]. It is defined by the presence of iridocorneal synechiae accompanied by variable density in the central corneal opacity (leukoma) [3]. Typically unilateral, PA1 exhibits a clear peripheral cornea and rarely shows signs of edema or scleralization [2,14,34]. Vision prognosis is generally favorable, and systemic abnormalities are uncommon in PA1 [3]. In contrast, PA2 arises from an incomplete separation between the cornea and lens, presenting with a more severe manifestation characterized by, i.e., corneolenticular synechiae directly adhering to the corneal opacity and/or corneal opacity and lens opacities as in cataracts [3,34]. PA2, as well as the later added terms PPS and PPLS, commonly affect both eyes. However, PA2 can be associated with systemic anomalies, whereas PPS and PPLS has to include them [22,34]. One may argue that a PA2 phenotype with systemic anomalies in today’s classification belongs to the PPLS subgroup. Most systemic abnormalities originated from primary non-ocular neural crest cells, which contribute to the development of cartilage, bone, connective tissues, components of teeth (excluding enamel), pigment cells, and the peripheral nervous system in the facial region [43]. PPS, previously known as Krause–Kivlin syndrome, is an autosomal recessive inherited congenital disorder defined by the identification of a pathogenic *B3GLCT* variant on each allele in addition to the presence of an ASD (typically PA) with additional systemic abnormalities (most commonly short stature and brachydactyly) [22,24,25,44]. PPLS presents with the same phenotype as PPS but lacks a homozygous or compound heterozygous *B3GLCT* variant, thus able to be diagnosed by exclusion in an inconclusive genetic workup or if other genetic variants are found [25,26,27]. In summary, PPS is the only subtype distinctly characterized by both genotypic and phenotypic features, whereas PA1, PA2, and PPLS exhibit only genotype–phenotype associations, thus suggesting overlapping boundaries among these subtypes. The terms PA2 in former research and PPLS in the current literature can be used interchangeably in some context due to evolving disease classifications. Initially, classifications were based on clinical features, but as genetic insights have advanced, more nuanced distinctions have emerged [45]. Table 1 provides an overview of disease classification across the literature, including disease-associated genes and inheritance modes described for PA spectrum disorder to date.

PA often leads to significant complications, including sensory deprivation, amblyopia, and glaucoma [35]. Sensory deprivation occurs due to corneal opacity obstructing the visual axis, resulting in reduced vision input and subsequent amblyopia [46]. The risk of developing amblyopia depends on the extent and localization of corneal opacity. Additionally, vision-threating glaucoma is a common complication, affecting approximately 30–70% of patients with PA due to the abnormal development of the trabecular meshwork and Schlemm’s canal [47]. Initiating prompt surgical treatment like peripheral iridotomy in clear peripheral cornea, penetrating keratoplasty in severely opaque cornea and/or lensectomy in vision-threatening cataracts, and/or anti-glaucomatous surgery in medically uncontrollable glaucoma may ensure better long-term vision outcomes in the management of PA spectrum disorder [35,48].

PA spectrum disorders mostly appear isolated and sporadic. Nevertheless, instances of both autosomal recessive and dominant, as well as x-linked, inheritance have been documented [10,49]. More comprehensive genomic testing as in whole-exome (WES) or -genome sequencing (WGS) is usually utilized in the genetic workup of PA spectrum disease [50].

This study presents a comprehensive genetic analysis of two unrelated patients with PA, identifying novel variants and assessing their clinical significance. By characterizing these genetic variants and their associated phenotypes, we offer new insights into PA spectrum disorder. The following sections outline our methodology, key discoveries, and their broader implications for ASD research.

## 2. Results

### 2.1. Case Presentation

Table 2 and Figure 1 provide an overview of the clinical characteristics and phenotypic features observed in the two patients diagnosed with PPLS and PA2. Both patients underwent thorough ophthalmological examination under general anesthesia (EUA) within the first six months of life, enabling detailed evaluation of anterior segment abnormalities and associated ocular findings. 

Patient P2 exhibited an isolated ocular phenotype, with no detectable extraocular anomalies during the initial and follow-up clinical assessments. In contrast, patient P1 presented with a more complex clinical picture. In addition to the ocular manifestations typical of PA, P1 also displayed multiple extraocular features, including distinct facial dysmorphism and a congenital heart defect.

#### 2.1.1. Patient 1

P1 was referred shortly after birth due to bilateral corneal opacities. At the initial examination at seven days of age, both corneas exhibited a bluish-white discoloration with limbal neovascularization, giving the eyes a smaller appearance. The AC was not visible. At three weeks of age, EUA revealed completely cloudy corneas with a ring-like structure (Figure 1). The corneal diameters measured 9 mm horizontally and 7 mm vertically, with an intraocular pressure (IOP) of 7 mmHg and an axial length of 19 mm in both eyes. Ultrasound biomicroscopy (UBM) identified anterior synechiae, a very shallow AC, and an indistinct pupil, with the anterior lens surface obscured by the iris. These findings confirmed a diagnosis of bilateral congenital ASD within the PA spectrum.

By two months of age, multiple surgical interventions, including penetrating keratoplasty or the use of a Boston keratoprosthesis combined with lensectomy, were considered following consultations with international experts in the field. However, these procedures were deemed high-risk due to the potential for glaucoma development, retinal detachment, and the need for multiple anesthesia sessions. Given the anticipated need for repeated surgical interventions, uncertain vision prognosis, and potential complications, the parents opted against surgical treatment. Regular IOP monitoring continued throughout childhood. The patient developed early-onset pendular nystagmus.

By age five, IOP remained within normal limits. UBM confirmed multiple iris adhesions to the endothelium, thickened corneas, and a small but present pupil, without lens-corneal adhesions or posterior segment abnormalities. As P1 developed, craniofacial anomalies became more evident, including small eyes, hypertelorism, a small midface with a prominent forehead, a short upper lip, a flat nasal bridge, and small, deep-set ears. Additionally, long fingers and an atrial septal defect were diagnosed, suggesting syndromic features. Currently, at thirteen years of age, P1 retains light perception, with stable ocular findings.

#### 2.1.2. Patient 2

P2 was referred as an emergency case by the neonatology due to bilateral absence of visible pupils. The initial ophthalmic examination revealed a small, displaced pupil with anterior synechiae and central corneal opacity in the right eye, whereas the left eye exhibited a central corneal opacity with no visible pupil (Figure 2). Mydriatic eye drops were prescribed for the right eye three times daily, and an urgent iridotomy was scheduled for the left eye.

At two months of age, an EUA recorded IOP values between 6 and 14 mmHg in both eyes, with corneal diameters of 10.5 mm (right) and 11 mm (left eye). Anterior segment findings included irregular iridocorneal synechiae and focal corneal opacities in both eyes, with complete pupil absence in the left eye. Fundoscopic examination revealed small but vital optic discs, absent macular reflexes, and an otherwise normal retina. UBM was not performed. A sectoral iridectomy was successfully performed on the left eye.

By three months of age, P2 exhibited wandering eye movements but maintained binocular fixation to light, with a stronger response from the left eye. Near fixation was mostly parallel, though alternating esotropia was observed. The right eye had a pharmacologically dilated pupil with synechiae, while the left eye showed an inferior iris defect after iridectomy, with a clear lens. The central corneal opacity remained unchanged in both eyes (Figure 2).

At six months, bilateral pendular nystagmus developed. EUA at seven months revealed stable findings including normal IOP. Retinoscopy confirmed high myopia (−7.0 diopters (D) in the right eye and −6.0 D in the left eye). Corneal diameters remained 11 mm bilaterally, with unchanged anterior segment findings. Fundoscopy revealed normally excavated optic discs with absent macular reflexes. Intermittent esotropia persisted, necessitating corrective lenses to support visual development, along with continued IOP monitoring and refraction assessments for appropriate spectacle correction. Topic mydriatic therapy was continued for the right eye until the age of three.

At twelve years of age, uncorrected visual acuity (VA) was 0.1 (right eye) and 0.05 (left eye). With correction, VA improved to 0.16 (right eye) and 0.1 (left eye). While IOP remained stable in the right eye, it increased in the left eye (22–27 mmHg). Anterior segment findings remained unchanged, with persistent iridocorneal synechiae, corneal opacity, and iris defect after iridotomy in the left eye. Ultrasound confirmed optic disc drusen later as the cause of disc elevation. Due to persistently elevated IOP, topical timolol therapy was initiated in the left eye. At thirteen years of age, VA remains stable in both eyes. IOP is well controlled in the left eye with pressure-lowering eye drops. Central visual field testing remains inconclusive due to erratic eye movements.

### 2.2. Segregation Analysis Results

Whole-exome sequencing (WES) was performed on P1 and P2 to identify potential disease-causing variants. As both were the only affected individuals in their families, variant filtering prioritized de novo and recessive biallelic variants, which are more likely to underlie their phenotype. Filtering criteria included a heterozygous allele frequency of ≤1% and a homozygous allele frequency of ≤0.01% in population databases, as well as a Combined Annotation Dependent Depletion (CADD) score ≥ 20. Variants classified as likely pathogenic (LP) or pathogenic according to the American College of Medical Genetics and Genomics (ACMG) guidelines were further prioritized.

Following bioinformatic filtering, Sanger sequencing was used to validate the identified variants, confirming them as true positives. The same sample was used, but with an alternative DNA extraction and sequencing approach to ensure the reliability of the findings and to exclude false positives. Similarly, segregation analysis of these variants was performed on the parents. No evidence of somatic mosaicism was detected, although low-level germline mosaicism in the parents could not be excluded. Importantly, neither parent carried the identified variants, supporting their de novo origin and strengthening their association with the observed phenotypes.

WES identified disease-related variants affecting dominant genes involved in essential biological processes. P1 harbors a heterozygous ~1.6 Mb deletion (chr8:g.76760782_ 78342600del) including genes *PEX2* and *ZFHX4*. P2 carries a likely pathogenic heterozygous *FOXC1* variant (NM_001453.3:c.310A>G), resulting in a p.Ile104Val substitution in exon 1. The results are summarized in Table 3.

Both variants are novel and have not been previously reported. Their potential role in disease warrants further investigation, particularly regarding their clinical significance and pathogenic mechanisms. These findings highlight the importance of comprehensive genetic screening in uncovering disease-associated variants in PA spectrum disorder.

### 2.3. Breakpoint Confirmation

To confirm the presence of the deletion involving the *PEX2* and *ZFHX4* gene loci, a combination of long-range PCR and Sanger sequencing was employed. A 5157 bp amplicon was successfully amplified, encompassing 3682 bp upstream of the deletion breakpoint and 1475 bp downstream. Long-range PCR was performed using TaKaRa LA Taq polymerase, ensuring robust amplification across the affected region.

Sequencing analysis identified a large genomic deletion spanning from position 76,760,781 to 78,342,601 (GRCh37), corresponding to a total deletion size of approximately 1.6 Mb on the long arm of chromosome 8 as visualized in Figure 3. Subsequent sequence alignment and depth-of-coverage analysis confirmed the absence of reads mapping to the deleted region in the WES data, further supporting the presence of a heterozygous deletion in the affected area.

To confirm the breakpoints identified with the long-range PCR, Sanger sequencing was conducted using primers flanking the deletion site. The sequencing results validated the breakpoints and confirmed the genomic deletion as chr8:g.76760782_78342600del in the GRCh37 reference genome. This comprehensive approach, integrating long-range PCR, sequencing analysis, and manual curation, ensured high-confidence identification and characterization of the structural variant.

The ~1.6 Mb deletion at chromosome 8q21.13 (chr8:g.76760782_78342600del) affects a contiguous genomic segment containing several protein-coding and non-coding elements.

The *PEX2* gene (https://www.ncbi.nlm.nih.gov/gene/5828, accessed on 13 April 2025), which comprises five exons and plays a critical role in peroxisomal biogenesis, is entirely located within the deleted interval and is therefore expected to be completely lost on the affected allele.

Similarly, the *ZFHX4* gene (https://www.ncbi.nlm.nih.gov/gene/79776, accessed on 13 April 2025), spanning eleven exons and involved in transcriptional regulation and neurodevelopmental processes, also lies fully within the deleted region and is predicted to be absent from the deleted allele.

In addition to these protein-coding genes, the deletion encompasses several long non-coding RNAs (*LINC01109*, *LINC01111*), as well as the microRNA gene *MIR3149*, which may have regulatory functions. The loss of these non-coding elements could contribute to altered gene regulation and phenotypic variability.

Structurally, the control allele retains a normal diploid configuration across this region, whereas the affected allele exhibits a heterozygous ~1.6 Mb deletion, resulting in a dosage loss of all genes and regulatory elements within this interval.

## 3. Discussion

In this study, we report two novel de novo variants in two unrelated patients diagnosed with PA spectrum disorder. The first patient harbors a heterozygous ~1.6 Mb interstitial deletion on chromosome 8q21.13 (GRCh37 chr8:g.76760782_78342600del) encompassing protein-coding genes *PEX2* and *ZFHX4*, while the second patient carries a heterozygous *FOXC1* missense variant (NM_001453.3: c.310A>G, p.Ile104Val).

Clinically, P1 exhibited circular complete corneal opacity, shallow AC, iridocorneal and lenticular synechiae, and systemic anomalies resembling a PPLS phenotype. P2 presented with central corneal opacity, a shallow AC, iridocorneal and lenticular synechiae, iris dysplasia, and optic nerve anomalies, consistent with a PA2 phenotype. These findings expand the known genetic heterogeneity of PA, emphasizing the involvement of both chromosomal deletions and single-gene variants in the ASD spectrum.

The *FOXC1* variant identified in P2 aligns with prior reports linking *FOXC1* variants to ASD, including Axenfeld–Rieger anomaly and PA spectrum disorders [11]. *FOXC1* encodes a forkhead (FH) transcription factor, which plays a critical role in neural crest-derived mesenchymal differentiation, forming the cornea, iris, and trabecular meshwork [51,52]. Pathogenic variants in *FOXC1* exhibit a continuum of phenotypes, from isolated Axenfeld–Rieger spectrum disorders to PA phenotype with severe corneal opacification and iridocorneal adhesions [16]. The identified *FOXC1* missense variant p.Ile104Val lies within the DNA-binding domain [53], a region essential for regulating downstream target genes, and likely disrupts transcriptional activity, leading to iridocorneal dysgenesis with secondary glaucoma [11]. This highly conserved isoleucine residue is located within the α-helical core of the FH domain, which is essential for proper nuclear localization and high-affinity DNA binding. Functional studies have shown that the FH domain contains two nuclear localization signals and is critical for directing *FOXC1* to the nucleus and enabling sequence-specific DNA interaction [54]. Although the p.Ile104Val substitution is conservative—replacing one large, hydrophobic residue with another—position 104 is evolutionarily invariant across vertebrates, as supported by a high PhyloP100way conservation score of 8.257 (according to https://varsome.com accessed on 23 June 2025) indicating strong purifying selection. Substituting isoleucine for valine at this critical position may lead to subtle but functionally significant alterations, such as local destabilization of helix packing or disruption of DNA contact surfaces. Even minor changes in this domain, particularly near residues involved in nuclear import, can result in protein mislocalization, impaired DNA binding, and decreased transcriptional activation, as demonstrated by Berry et al., 2002 [54]. Consistent with this, PolyPhen-2 predicts the variant to be probably damaging with a high score of 0.996 (sensitivity: 0.55; specificity: 0.98) (according to http://genetics.bwh.harvard.edu/pph2/ accessed on 10 June 2025). Although no functional assays have been reported specifically for p.Ile104Val, the p.Ile104Thr variant has been classified as a variant of uncertain significance in ClinVar (ID: 2858932), despite its segregation in individuals exhibiting features of *FOXC1*-related conditions. Additionally, the nearby variant p.Phe112Ser has been reported as pathogenic in a family affected by Axenfeld–Rieger syndrome [11]. Taken together, this underscores the sensitivity of this domain. Furthermore, the variant’s position within a functionally essential region of the FH domain and the known intolerance of this area to structural changes support its pathogenicity. Functional dissection studies confirm that the domain’s nuclear localization signals are vulnerable to small secondary structure disruptions [54], and prior mutagenesis shows nearby substitutions impair *FOXC1*’s nuclear import and DNA binding [53,54]. Thus, despite the lack of direct assay data, the combination of conservation, critical domain context, neighboring pathogenic variants, and computational predictions supports a deleterious effect. This aligns with ACMG classification criteria (PM1, PM2, PP2, PP3, and PS2) and justifies interpretation of this variant as likely pathogenic. Prior studies have shown that both haploinsufficiency and gain-of-function effects contribute to the phenotypic variability observed in *FOXC1*-related disorders, with dosage sensitivity playing a critical role in disease expression [54,55]. In our case, the presence of IOP elevation in P2 is consistent with *FOXC1* dysfunction, which disrupts trabecular meshwork and Schlemm’s canal formation, leading to impaired aqueous humor outflow [11].

The *PEX2-ZFHX4* deletion in P1 aligns with previous reports of 8q21.11 microdeletions in PA spectrum cases presenting with systemic anomalies [15,55,56,57]. *ZFHX4* encodes a large zinc-finger homeobox transcription factor implicated in ocular, craniofacial, and neurodevelopmental processes [55]. It is highly expressed in early embryonic eye structures, and haploinsufficiency disrupts mesenchymal–epithelial interactions, leading to AC malformations [15]. Patients with *ZFHX4* deletions commonly exhibit variable ASD phenotypes, ranging from isolated corneal opacities to syndromic ASD with craniofacial and systemic involvement [15,55]. In our patient, the combination of hypertelorism, midface hypoplasia, and congenital heart defects is consistent with this broader PPLS phenotype [15,56,58]. While *ZFHX4* loss is likely the primary driver of the ASD and craniofacial anomalies, the co-deletion of *PEX2* may act as a phenotypic modifier. *PEX2* encodes a peroxisomal membrane protein essential for peroxisome biogenesis and metabolic homeostasis [59,60]. Biallelic *PEX2* variants cause Zellweger spectrum disorders, which frequently include hypotonia, liver dysfunction, developmental delay, and ocular abnormalities such as corneal opacities, retinal anomalies, and AC malformations [15,61,62,63]. Although heterozygous loss of *PEX2* has not been functionally characterized, the essential role of the gene and the severity of biallelic phenotypes suggest that even partial loss could contribute to disease expression in a dosage-sensitive manner [63,64]. In our case, the presence of systemic features—including long fingers and a congenital heart defect—may reflect such a modifying effect [62,64]. This hypothesis is further supported by prior reports of 8q21.11 microdeletions where *PEX2* and *ZFHX4* are co-deleted, and where patients consistently presented with syndromic ASD resembling PPLS [15,55]. Taken together, we propose that *ZFHX4* haploinsufficiency is responsible for the core ocular and craniofacial phenotype, while *PEX2* loss likely contributes to the multisystem features observed in our patient.

Our findings highlight the importance of comprehensive genetic testing in PA spectrum disorders. Standard gene panel testing may miss large deletions, emphasizing the necessity of CNV analysis in ASD cases that remain without a molecular diagnosis. Identifying the *PEX2*-*ZFHX4* deletion in P1 provides a valuable diagnostic marker, warranting multidisciplinary surveillance, including cardiac and neurodevelopmental monitoring [15,23]. These findings also enhance genetic counseling, as de novo variants indicate a low recurrence risk, although germline mosaicism cannot be entirely excluded.

## 4. Materials and Methods

### 4.1. Patients

The identification of the two unrelated index patients was achieved through a genetic study centered on ASD like PA. This study, conducted by the Department of Ophthalmology at University Hospital Zurich in collaboration with the Institute of Medical Molecular Genetics at the University of Zurich, seeks to map out the phenotypic and genotypic features of PA. Both patients included received an eye examination including a dilated fundus examination, and extraocular manifestations were documented thoroughly. Comprehensive retrospective chart reviews were carried out. Blood samples were drawn from the patients and their biological parents. This study complied with Good Clinical Practices and adhered to the Declaration of Helsinki guidelines. The Cantonal Ethics Committee of Zurich (Ref-No. 2019-00108) approved the genetic testing of human patients. Written consent was obtained from the patients’ legal guardians.

### 4.2. Gene Targets

The extensive gene catalogue by Lang et al. (2020) [65] containing ASD and congenital glaucoma-associated genes known up to 2020 was expanded using the Human Gene Mutation Database (HGMD) along with an updated extensive literature search of newly discovered associated genes (Appendix A).

### 4.3. Exome Data Sequencing and Review

We conducted whole-exome sequencing (WES) and sequence data analysis following established protocols [65,66,67,68]. In summary, DNA was extracted from EDTA blood samples using the Chemagic DNA Blood Kit (Perkin Elmer, Waltham, MA, USA), and the DNA was fragmented using an M220 Sonicator (Covaris, Woburn, MA, USA). Libraries were prepared according to the IDT-Illumina TruSeq DNA exome protocol (Illumina, San Diego, CA, USA, and Integrated DNA Technologies, Coralville, IA, USA). Paired-end sequencing (2 × 75 bp) was performed on a NextSeq 550 system (Illumina, San Diego, CA, USA). The sequencing reads were aligned to the human genome reference (GRCh37) using Burrows–Wheeler Aligner (BWA) v0 m.7.17 on BaseSpace Onsite (Illumina). Variant annotation was performed using the SOPHiA DDM™ Platform and Alamut™ Visual Plus (SOPHiA GENETICS, Lausanne, Switzerland). Copy number variations (CNVs) within the genes of interest (Appendix A) were identified from exome coverage depth data using Sequence Pilot version 5.0 (JSI Medical Systems GmbH, Ettenheim, Germany). Variants with a heterozygous allele frequency greater than 1% and a homozygous allele frequency above 0.01% (according to gnomAD; https://gnomad.broadinstitute.org/ accessed on 8 June 2024) were excluded.

### 4.4. Segregation Analysis

We carried out segregation analysis using Sanger sequencing following the detailed methodology outlined by Haug et al. (2021) [67]. Briefly, PCR was used to amplify the target regions. PCR products underwent cycle sequencing using BigDye™ Terminator V1.1 (Thermo Fisher Scientific, Waltham, MA, USA). Post-sequencing, the products were purified, and the sequencing was executed on a SeqStudio capillary sequencer (Thermo Fisher Scientific, Waltham, MA, USA).

### 4.5. Breakpoint Assessment

To confirm the deletion involving the *PEX2* and *ZFHX4* gene loci, long-range PCR and next-generation sequencing (NGS) were employed as previously described [69]. Two sets of primers were designed to amplify regions flanking the suspected deletion breakpoints, based on the human genome assembly (GRCh37). For the first set, the forward primer (Primer 1F) was positioned at nucleotide 76757099, and the reverse primer (Primer 1R) was positioned at nucleotide 76760781. For the second set, the forward primer (Primer 2F) was located at nucleotide 78342601, and the reverse primer (Primer 2R) was located at nucleotide 78344076. These primer positions correspond to specific base pair locations on chromosome 8, with Primer 1F and Primer 1R flanking the *PEX2* locus and Primer 2F and Primer 2R flanking the *ZFHX4* locus.

To detect and characterize the deletion, Primer 1F (upstream of *PEX2*) and Primer 2R (downstream of *ZFHX4*) were used together to amplify across the deletion breakpoints, producing a fragment only if the deletion was present. Primers flanking the suspected breakpoint were designed by comparing exome sequencing data upstream and downstream of the deleted genes with sequencing data from a control sample. Conversely, Primer 2F and Primer 1R were used to amplify a region within the suspected deletion; failure to produce a product would indicate the presence of the deletion, as indicated in Figure 2.

The PCR conditions included an initial denaturation at 94 °C for 2 min, followed by 35 cycles of 98 °C for 10 s and 68 °C for 12 min, with a final extension at 72 °C for 10 min. The resulting PCR products were fragmented to an average size of 350 bp using a Covaris M220 ultrasonicator. Library preparation was carried out with the TruSeq DNA Nano Kit, quantified using Qubit, and quality-checked with the Agilent High-Sensitivity DNA Kit. Sequencing was performed on the MiSeq system (Illumina). Reads were aligned to the hg19 reference genome using BWA-MEM and analyzed with GATK.

To enhance confidence in the breakpoint assignment, the precise fusion junction was subsequently validated by Sanger sequencing. Specific primers flanking the predicted junction were designed, and bidirectional sequencing confirmed the exact breakpoint location. To further investigate the inheritance pattern of the deletion, segregation analysis in family members was performed using multiplex PCR, followed by agarose gel electrophoresis and analysis on a 2100 Bioanalyzer.

## 5. Conclusions

Our study identifies two novel de novo variants in unrelated patients, expanding the known genetic landscape of PA spectrum disorders. The first case involves a de novo ~1.6 Mb interstitial deletion on chromosome 8q21.13 (GRCh37 chr8:g.76760782_78342600del) affecting the *PEX2* and *ZFHX4* genes, while the second patient carries a heterozygous *FOXC1* missense variant (NM_001453.3:c.310A>G). These findings highlight the genetic heterogeneity of PA spectrum disorders and provide new insights into their clinical manifestations.

## Figures and Tables

**Figure 1 ijms-26-06454-f001:**
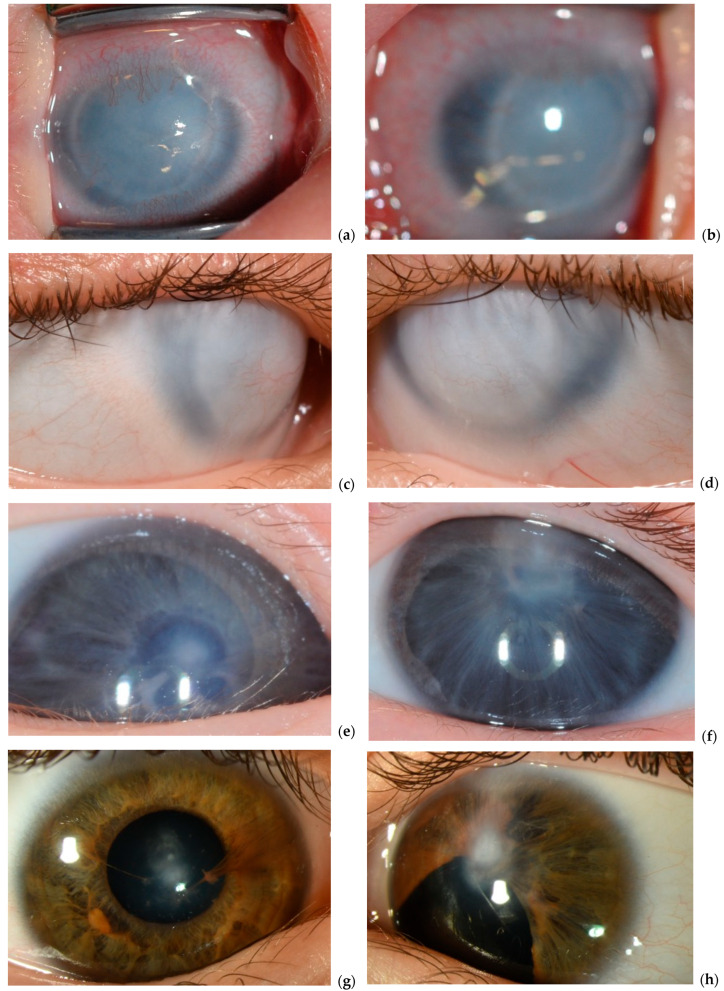
Slit lamp photos of initial and follow up clinical manifestation. Right eye (**a**) and left eye (**b**) at two months and right (**c**) and left (**d**) eye at six years of age in P1. Right eye (**e**) and left eye (**f**) at two months and right eye (**g**) and left eye (**h**) at eleven years of age in P2.

**Figure 2 ijms-26-06454-f002:**
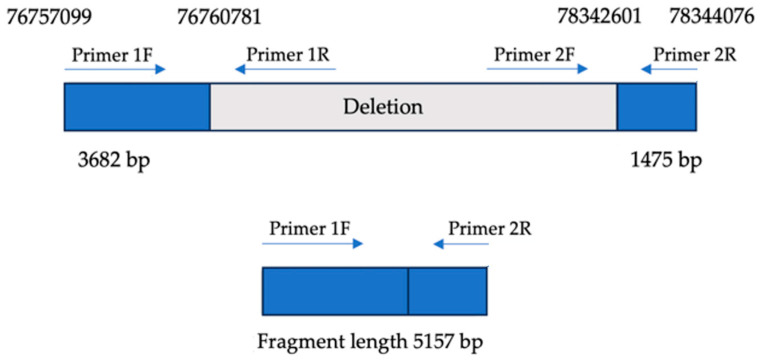
Breakpoint mapping. Upper panel: genomic organization of the ~1.6 Mb deletion on chromosome 8q21.13, encompassing the *PEX2* and *ZFHX4* loci. Blue boxes represent non-deleted flanking regions; gray indicates the deleted interval. Primer pairs (1F/1R and 2F/2R) were designed to amplify upstream and downstream regions flanking the deletion. Lower panel: long-range PCR using primers 1F and 2R yields a 5157 bp fragment only in the presence of the deletion, confirming the fusion of non-contiguous genomic regions.

**Figure 3 ijms-26-06454-f003:**
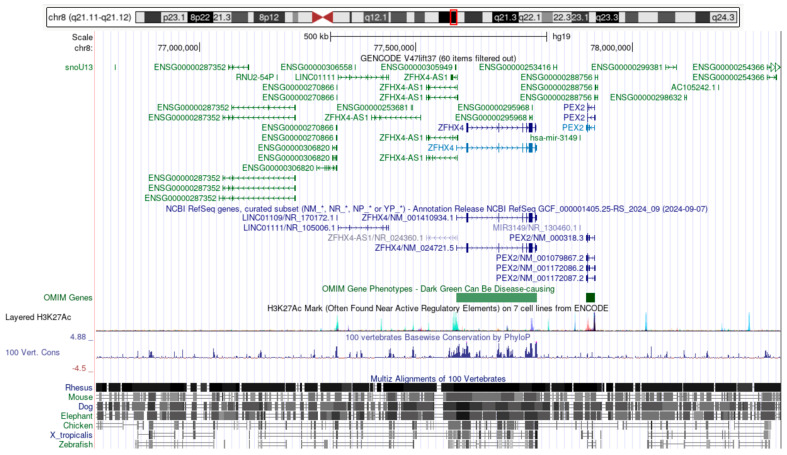
Genes affected by the ~1.6 Mb deletion at chromosome 8q21.13 (GRCh37: chr8: g.76760782_78342600del). The schematic illustrates the ~1.6 Mb deletion on chromosome 8q21.13 (GRCh37: chr8:g.76760782_78342600del), highlighting the affected genomic region and its gene content. The deletion spans several coding and non-coding genes, including *PEX2*, *ZFHX4*, and *MIR3149*, as well as multiple long non-coding RNAs. Evolutionary conservation and regulatory marks suggest the functional relevance of this region. (Figure derived from https://genome.ucsc.edu/cgi-bin/hgGateway?db=hg19, accessed on 13 April 2025).

**Table 1 ijms-26-06454-t001:** Peters anomaly spectrum disease classification: a phenotype–genotype overview in the literature.

	Peters Anomaly Type 1 (PA1)	Peters Anomaly Type 2 (PA2)	Peters Plus Syndrome (PPS)	Peters Plus-Like Syndrome (PPLS)
Associated genes (mode of inheritance)	*CDH2* (ad) [5], *COL4A1* (ad) [6], *COL6A3* (ad/ar) [7], *CYP1B1* (ar) [8], *DOP1B* (ar) [9], *FLNA* (xld) [10], *FOXC1* (ad) [11], *FOXE3* (ad/ar) [12], *HCCS* (xld) [13], *NDP* (xlr) [10], *PAX6* (ad) [14], *PEX2* (ar) [15], *PITX2* (ad) [16], *PITX3* (ad) [10], *PXDN* (ar) [17], *SLC4A11* (ar) [10], *SOX2* (ad) [18], *TFAP2A* (ad) [19], *ZFHX4* (ad) [15]		*B3GLCT* (ar) [20,21,22,23,24]	*absent B3GLCT* (ar) variant [25,26,27]; no genetics defined
Characteristic congenital ocular findings	Corneal abnormality (i.e., central corneal opacity plus defect or absent posterior stroma. Descemet’s membrane, endothelium and/or iridocorneal adhesion) [2,28,29,30,31]	Corneal abnormality (i.e., central corneal opacity plus defect/absent posterior stroma, Descemet’s membrane and/or endothelium) and lens abnormality (i.e., cataract and/or corneolenticular and/or iridolenticular adhesion and/or aphakia and/or lens remnant) [2,28,29,30,31]	ASD in 98% (thereof 73% with PA type 1 and 2 (distribution unclear), 25% with Axenfeld–Rieger spectrum, sclerocornea, and/or posterior embryotoxon), and 2% with other congenital ocular abnormalities) [22,23,32,33]	ASD (PA type 1 and 2), Axenfeld–Rieger spectrum) [25,33]
Laterality	Predominantly unilateral occurrence [2,34,35]	Predominantly bilateral occurrence [2,34,35]		
Optional congenital ocular findings	Congenital glaucoma, microphthalmia, PFV, iris and/or retina coloboma and/or hypoplasia, vitreal and/or retinal dysplasia, optic nerve and/or optic chiasm dysplasia, congenital anterior staphyloma, congenital myopia, ptosis [4,30,32,36,37,38,39,40,41,42]			
Secondary ocular findings	Glaucoma [29,32]			
Systemic and/or dysmorphic findings	Generally uncommon [34]	Generally uncommon, can include: Growth retardation, cleft/lip palate, facial dimorphism, brachycephaly, cardiac defects, neural defects, hearing deficits described in individual cases [35]	Must include either one: growth retardation, dwarfism, clinodactyly dig. 5, brachydactyly, developmental delay, hypopituitarism, central nervous system defects, cleft/ lip palate, congenital heart defect, genitourinary defect, cupid bow upper lip, long philtrum, narrow palpebral fissures, hypertelorism, broad neck, prominent forehead, micrognathia, small ears, pre-auricular pits, microcephaly, macrocephaly [4,23,25,32]	

Abbreviations: ad: autosomal dominant; ar: autosomal recessive; xld: x-linked dominant; xlr: x-linked recessive; ASD: anterior segment dysgenesis; PFV: persistent fetal vasculature.

**Table 2 ijms-26-06454-t002:** Clinical attributes, manifestation, and diagnosis of index patients.

ID	P1	P2
Sex	female	male
Suspected diagnosis	PPLS	PA2
Ocular findings at 2 months of age	Normal IOP, circular complete corneal opacity (i.e., sclerocornea), shallow AC, iridocorneal synechiae, iridolenticular synechiae, unremarkable posterior segment (B-scan ultrasound)	Normal IOP, central corneal opacity, shallow AC, iridocorneal synechiae, iridolenticular synechiae, localized lens opacity at iridolenticular adhesion site, iris dysplasia, optic nerve anomaly
2ry ocular findings	Pendular nystagmus	Elevated IOP, high myopia, pendular nystagmus, esotropia, optic nerve head drusen
Dysmorphic and/or systemic findings	Small eyes, hypertelorism, small midface, prominent forehead, thin vermilion border, short upper lip frenulum, smooth elongated philtrum, flat nose bridge, small deep-set ears, long fingers, atrial septal defect.	None
VA (Snellen decimal)	LP (BE) ^1^	0.16 (RE); 0.1 (LE) ^1^

Abbreviations: AC: anterior chamber; IOP: intraocular pressure; VA: visual acuity; LP: light perception; BE: both eyes; RE: right eye; LE: left eye. ^1^: at 13 years of age.

**Table 3 ijms-26-06454-t003:** Disease-causing variants identified by WES.

ID	Gene (Inheritance)	Gene Function (https://www.omim.org accessed on 9 December 2024)	Reference Sequence	Sequence Variant	Predicted Protein Change	Region/ Size	GnomAD	Zygo-sity	ACMG	CADD
P1	*PEX2* (ar)	Peroxisome biogenesis, lipid metabolism, detoxification	GRCh37	chr8:g.76760782_78342600del ^1^	n/a	~1.6 Mb	n/a	het	P	n/a
	*ZFHX4* (ad)	Transcription factor, cell differentiation and proliferation								
P2	*FOXC1* (ad)	Transcription factor, regulation of eye development, cell migration	NM_001453.3	c.310A>G ^1^	p.Ile104Val	exon 1	n/a	het	LP	33

Abbreviations: GnomAD: Genome Aggregation Database; ACMG: American College of Medical Genetics and Genomics; CADD: Combined Annotation Dependent Depletion; ar: autosomal recessive; ad: autosomal dominant; GRCh37: human genome assembly; n/a: not applicable; het: heterozygous; LP: likely pathogenic; ^1^: de novo.

## Data Availability

Relevant data from this study can be obtained upon request.

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
