# Peer review of "Novel Genetic Variants and Clinical Profiles in Peters Anomaly Spectrum Disorders"

_ijms, 2025, doi:10.3390/ijms26136454_

Round 1
Reviewer 1 Report
Comments and Suggestions for Authors
Thank you for the opportunity to review "Novel Genetic Variants and Clinical Profiles in Peters Anomaly Spectrum Disorders." This study identifies and characterizes novel de novo genetic variants in two unrelated individuals with Peters Anomaly (PA) spectrum disorder using whole-exome sequencing and further molecular techniques. The authors report a ~1.6 Mb deletion encompassing PEX2 and ZFHX4 in a patient with Peters-Plus like syndrome (PPLS) and a likely pathogenic FOXC1 missense variant (c.310A>G) in a patient with isolated PA type 2. These findings aim to broaden the understanding of the genetic heterogeneity underlying these complex anterior segment dysgenesis disorders.
The manuscript contributes to the field of ophthalmic genetics by detailing new genetic causes for a rare and challenging group of conditions. The methodological approach to variant identification and confirmation, including breakpoint analysis for the deletion, is generally robust, and the clinical phenotypes are reasonably well-described. The clear presentation of two distinct cases with novel genetic findings highlights the complexity of the PA spectrum disorders. It underscores the utility of comprehensive genomic investigation in their diagnosis and classification.
While promising, several areas require clarification to strengthen the manuscript's conclusions and impact. Primarily, the novelty and predicted functional consequences of the FOXC1 p.Ile104Val variant need to be more rigorously substantiated beyond current database checks and CADD scores. Further elaboration is also required on the specific contribution of PEX2 haploinsufficiency to the systemic features observed in the PPLS patient, distinguishing its role from that of the co-deleted ZFHX4. Additionally, minor clarifications are required regarding the breakpoint analysis illustration (Figure 1) and the precise workflow for breakpoint confirmation (NGS followed by Sanger) to enhance methodological transparency. Addressing these major points, along with other minor textual and referencing details, will significantly improve the manuscript. Therefore, I recommend Major Revisions.
Reviewer 2 Report
Comments and Suggestions for Authors
This manuscript reports two unrelated cases of Peters anomaly (PA) spectrum disorders identified via whole-exome sequencing (WES). In one patient, the authors identified a novel heterozygous ∼1.6 Mb deletion spanning PEX2 and ZFHX4, while the other harbored a heterozygous FOXC1 missense variant (c.310A>G, p.Ile104Val). The study includes detailed clinical phenotyping, genetic analysis, and contextualizes the findings within the broader framework of anterior segment dysgenesis (ASD). The manuscript is well-organized and clearly written.
-
In Table 2, both "AC" and "anterior chamber" are used. Please standardize the terminology for clarity and consistency.
-
The discussion of the FOXC1 missense variant (p.Ile104Val) would benefit from additional detail. Please elaborate on its potential functional implications, including its location within the protein domain and predicted impact on protein function.
Round 2
Reviewer 1 Report
Comments and Suggestions for Authors
Thank you for the opportunity to review the manuscript entitled, "Novel Genetic Variants and Clinical Profiles in Peters Anomaly Spectrum Disorders". The revision is well-structured and Delas et al. present a comprehensive genetic and clinical analysis of two unrelated patients with Peters Anomaly (PA) spectrum disorder.
Through whole-exome sequencing (WES) and meticulous validation, the authors identified two novel, de novo variants . The first patient, presenting with a Peters-Plus like syndrome (PPLS) phenotype, was found to harbor a heterozygous ~1.6 Mb deletion on chromosome 8q21.13, which encompasses the PEX2 and ZFHX4 genes . The second patient, with a PA type 2 phenotype, carries a likely pathogenic missense variant in FOXC1 (p.Ile104Val) .
This manuscript is a high-quality piece of research that is thorough, well-executed, and clearly presented. It meets the standard for publication in the International Journal of Molecular Sciences. I recommend accepting without revision & I am grateful for the opportunity to review this well-executed work.